# Weakly-supervised explainable infection severity classification from chest CT scans

**Ibrahim Almakky**[ID]*, **Mohammad Yaqub**

Mohamed Bin Zayed University of Artificial Intelligence, Abu Dhabi, United Arab Emirates

* ibrahim.almakky@mbzuai.ac.ae

**Data availability statement:** As specified within the manuscript the datasets used in this work are available from: 1. Ning W, Lei S, Yang J,

## Abstract

Novel respiratory diseases can have a devastating impact on healthcare systems, which underlines the importance of developing methods to improve the prevention, diagnosis, and prognosis of such diseases. Developing computer-aided diagnosis tools that determine infection severity can aid healthcare professionals in deciding treatment strategies and preventing cross-infection. In such manner, lung infection severity classification from chest CTs is crucial for deciding treatment plans and interventions needed to block illness progression in individual patients and reduce cross infection. However, current techniques face performance, generalizability, and explainability challenges for automated infection severity classification methods due to the high spatial complexity of 3D volumes. Significant efforts have been focused on segmentation approaches to quantify lung infection and assess infection severity, but such segmentation methods require expensive data annotation and clinical expertise. In this work, we propose a weakly-supervised classification approach to distinguish between different levels of infection, while providing clinicians with explainable results. To mimic clinical practice, the different stages in our approach focus on low-level infection patterns as well as high-level infection coverage in lung CT scans. We then fuse the high-level features with the positionally encoded low-level features to provide volume-level infection classification. Testing on the SARS-CoV-2 (COVID-19) multi-centre multi-region datasets, our approach shows promising performance gains compared to existing state-of-the-art methods, where we achieve state-of-the-art severity classification performance. Furthermore, we show significant performance gains on cross-site train/test splits. Finally, we quantitatively and qualitatively demonstrate the explainability of our weakly-supervised approach, where we can achieve substantial infection coverage.

## Introduction

The outbreak of novel respiratory diseases poses a great risk to healthcare systems around the globe. With long incubation periods and high transmissibility, such diseases can have more adverse effects on society. COVID-19 was the most recent use case, where infections and the resulting strain on healthcare institutions worldwide made it clear that Artificial Intelligence (AI)-assisted screening and diagnosis can alleviate some of this strain. With this in mind, researchers around the world set out to develop deep learning models to assist with the

Cao Y, Jiang P, Yang Q, et al. Open resource of clinical data from patients with pneumonia for the prediction of COVID-19 outcomes via deep learning;4(12):1197–1207. https://doi.org/10.1038/s41551-020-00633-5. 2. Morozov SP, Andreychenko AE, Pavlov NA, Vladzymyrskyy AV, Ledikhova NV, Gombolevskiy VA, et al. MosMedData: Chest CT Scans with COVID-19 Related Findings Dataset; p. 2020.05.20.20100362. https://doi.org/10.1101/2020.05.20.20100362.

**Funding:** The author(s) received no specific funding for this work.

screening and diagnosis of COVID-19 infections. The aim is to support the medical community in curbing the spread of the virus and to manage the treatment for those infected. Reverse Transcription Polymerase Chain Reaction (RT-PCR) tests have been considered as the gold standard for COVID-19 diagnosis. However, X-ray and CT scans have been used to complement RT-PCR tests for diagnosis, follow-up assessment, and monitor disease evolution.

The first priority was to employ Deep Learning models to aid the diagnosis of COVID-19 from chest X-ray images, where many approaches have been proposed to distinguish between normal, pneumonia, and COVID-19 cases [1–4]. However, infected patients can experience varying levels of clinical manifestations, from no symptoms to critical illness [5]. Thus, identifying the severity of the infection is important to determine intervention and treatment strategies, which can vary from telemedicine in mild cases to intensive care in critical cases. While X-ray images can be used for initial COVID-19 diagnosis, CT imaging can provide a better view to determine the infection severity. The characteristics of COVID-19 in CT images reportedly include patchy Ground-Glass Opacities (GGOs) and large consolidation in the peripheral of both lungs [6]. Furthermore, the severity of the COVID-19 infection is related to the size and type of these abnormalities. However, the definition of COVID-19 infection severity varies between guidelines defined in different countries, but they share key elements when it comes to the disease progression. To achieve better generalizability, we work with datasets from different countries, where the infection severity was labelled according to two different guidelines. Some work has been done to classify COVID-19 infection severity from multi-center settings [7,8], but they are limited to a single region and unified label definitions.

CT images along with clinical features have been employed to classify COVID-19 infection severities and to predict patient outcomes [7]. However, the accuracy, generalizability, and explainability of such methods still face challenges due to the complexity of CT volumes and the subtle visual differences between the varying levels of infection. Explainability in particular, is a crucial component for diagnosis methods and tools to be considered for clinical adoption. Such explainability can be achieved through segmentation models, which can quantify infection in chest CT scans. However, manual annotation of lung infections is a tedious and time-consuming task, making it difficult to collect enough data to train a reliable infection segmentation model. In this work, we propose a novel weakly-supervised approach to provide an explainable infection severity classification from chest CT scans. We focus on low-level infection features as well as high-level infection coverage in the lungs to better differentiate between different infection severities. As such, this paper's contributions can be summarized as follows:

- To mimic clinical practice, we propose an end-to-end multi-stage pipeline to extract features of low-level infection patterns as well as high-level infection coverage within the CT volume.
- We develop a weakly-supervised approach to train the multi-stage pipeline to classify infection severities and to highlight infection regions for explainability.
- We propose a novel proximity factor between local infection clusters to encourage more accurate and faster-converging infection localisation.
- We demonstrate the generalizability of the proposed method using two datasets from two countries that follow different diagnostic guidelines and make the unified dataset and labels publicly available.

The remainder of this paper is organized as follows: In Sect 1, we describe work related to COVID-19 infection severity classification and segmentation, while also describing similar weakly-supervised approaches developed for 2D and 3D modalities. Following that in Sect

2, we discuss our proposed methodology in detail after which in Sect 3, we discuss details of the experimental setup including the datasets used and model architectures. Then in Sect 4, we report the test results of our approach and compare them to other state-of-the-art methods. In Sect 5 we delve into a discussion surrounding the results and methodology before we finally end with the conclusion and future work in Sect 6. As part of this work, we also release the code, trained models and combined datasets (https://github.com/BioMedIA-MBZUAI/InfectionSeverity-LungCT.git).

# 1 Related work

This section describes existing work on automated diagnosis and infection severity classification of COVID-19, infection quantification using segmentation models, as well as weakly supervised methods.

## 1.1 COVID-19 diagnosis

In the fight against the COVID-19 pandemic, enormous effort was dedicated to employ Deep Learning models for more accurate medical image analysis for both X-ray and CT images. With low radiation and low cost, chest X-ray has been widely leveraged to provide initial assessment of COVID-19 infections. Therefore, many models, specifically deep 2D CNNs, have been developed to classify and detect COVID-19 infections from chest X-ray images [9–12].

Chest CT, on the other hand, is a key part of the diagnostic process for suspected COVID-19 infections [13]. Previous research efforts on COVID-19 diagnosis from chest CT images have approached the task as either classification [7,14–16] or segmentation [17–19]. The classification task is employed to either distinguish between normal, pneumonia, and COVID-19 cases or to determine the severity of COVID-19 infection. Segmentation, on the other hand, is focused on infection quantification within the lungs. Ardakani et al. [14] conducted a comparison between ten common 2D deep CNN architectures and achieved the best performance using ResNet-101 and Xception models. Similarly, Gozes et al. [15] used an ImageNet pretrained ResNet-50 model to classify positive COVID-19 cases based on 2D image slices from input CT scans. Furthermore, Ning et al. [7] developed a dual Deep CNN architecture with a simplified VGG-16 model to first classify positive for COVID-19 CT slices, which are then fed into a secondary VGG-16 model to determine the severity of the case. More recently, Mondal et al. [16] proposed a Vision Transformer (ViT) classification model to screen COVID-19 from chest CT and X-ray images. Slice-based approaches ignore important 3D spatial relationships and are not inherently designed for explainability, while ViT models require a large amount of training data that is difficult to acquire in the medical domain. In contrast, 3D convolutional models allow for 3D spatial features to be considered during classification. In such manner, Xu et al. [20] trained a 3D DenseNet model to classify the CT volumes into four classes: healthy, bacterial pneumonia, COVID-19, or other viral pneumonia.

## 1.2 Explainability

Explainability is crucial for diagnostic methods to be considered for clinical adoption [21]. As a result, some works have focused on explainability using gradient-based approaches from classification CNNs [15] or from ViT models [16]. On the other hand, segmentation models are explainability-focused, where the model is trained to segment and quantify infection in the lungs. However, manual delineation of infection regions in CT images is a tedious task, labour-consuming, expensive, and challenging due to the highly varied textures, sizes, and

shapes. Wu et al. [18] collected a dataset of 3,855 CT image slices from 200 patients and annotated fine-grained pixel-level labels of lung opacifications. They used this dataset to jointly train classification and segmentation models to classify the image and segment lung infection. Zhou et al. [22] approached explainability from a different perspective. They proposed an ensemble of Deep Learning-based segmentation models to exclude tissue irrelevant to the pulmonary parenchyma from the CT volume. Then, following a human-in-the-loop training scheme, they extract interpretable subvisual CT lesions. Formulating the task as an infection detection, one can slightly reduce the annotation workload. For example, Ji et al. [23] proposed a single-stage model for automatic lung tumor detection in CT images, but the need for expert-level detection labels is still present.

### 1.3 Weak supervision

In 2D natural scene images, Zhou et al. [24] explored weak-supervision for instance segmentation with image-level labels. They exploit class peak responses in CNN models to enable instance mask extraction. In a similar manner, Ou et al. [25] leverage local peaks in image-level attention along with relationships between them to infer pseudo labels. For COVID-19 diagnosis, tackling annotation scarcity for infection regions, Han et al. [8] proposed a weakly-supervised attention-based multiple instance learning (MIL) approach for COVID-19 screening from chest CT volumes. This involves generating 3D instances using a deep instance generator, which are then passed to an attention-based MIL pooling to combine instances into a single representation. Converting the 3D CT scan into a bag of instances allows for the attention-based mechanism to provide insight into the contribution of each instance to the bag class label, and thus providing an interpretable output.

## 2 Methodology

This work aims to accurately classify infection severity from visual features in CT scans, while attaining explainable predictions through a weakly-supervised training approach. The end-to-end classification pipeline summarized in Fig 1 first takes the input volume through a cuboid nominator which picks "useful" cuboids. Selected cuboids are then passed through a cuboid-level binary classifier trained using pseudo-labels. The representation of the overall volume from the cuboid nominator along with the cuboid-level representations are finally passed to a fully-connected model tasked with volume-level classification. The remainder of this section describes each of those components in detail.

### 2.1 Proposal generator model

The proposal generator is composed of two components: (a) A cuboid nominator, which is a 3D CNN model that carries out the cuboid nomination. (b) A cuboid classifier, which is a small 3D CNN model tasked with cuboid-level classification.

**2.1.1 Cuboid nomination.** We conduct 3D patchification of the input volume $V$ of size $(z \times h \times w)$ dividing it into a set of $K$ non-overlapping cuboids, each of size $(\frac{z}{K} \times \frac{h}{K} \times \frac{w}{K})$. The cuboid nomination model, a 3D CNN, takes $V$ as input and is tasked with proposing $k$ "useful" cuboids for the overall classification of $V$, where $k \leq K$. This Cuboid Nominator is trained to classify each cuboid into two classes: "useful" or not. A cuboid is then considered "useful" if it leads along with other selected cuboids to an overall correct classification. As the task of nominating cuboids is simpler than learning global and regional visual features from the overall volume, a relatively small 3D CNN model can be used.

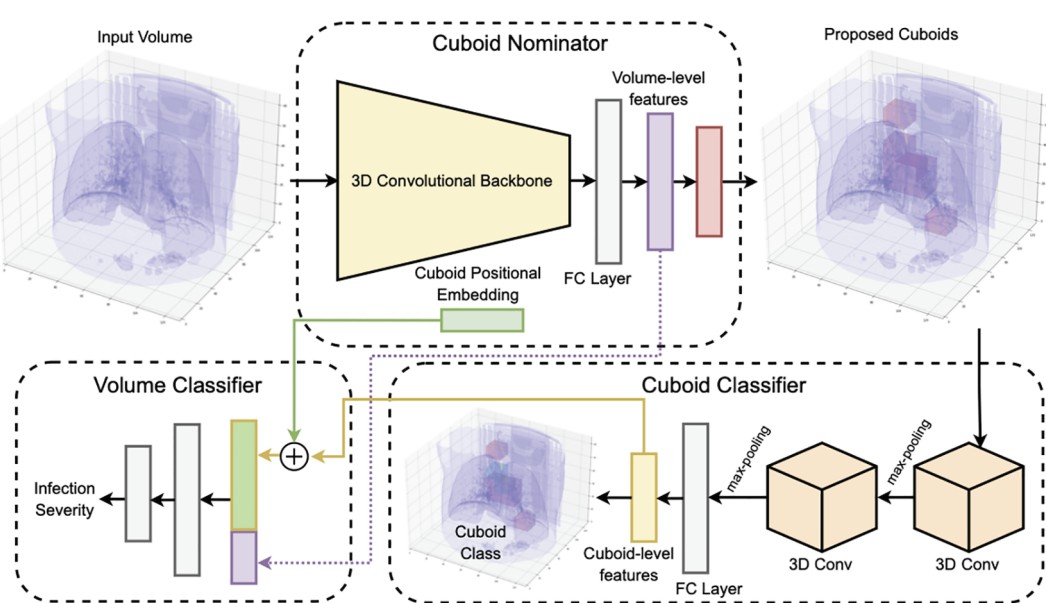

**Fig 1. Our proposed approach showing the role of the cuboid nominator in selecting cuboids from the input CT scan, while also extracting meaningful volume-level representation.** The selected cuboids are then fed into a cuboid classifier, which extracts cuboid-level features. The volume-level representation is then concatenated with the sum of the cuboid-level representation and their corresponding positional embedding to provide the patient-level classification.

**2.1.2 Cuboid-level classification.** The $k$ nominated cuboids are extracted from $V$ and fed into the Cuboid Classifier, which is a small 3D CNN model tasked with classifying each cuboid as either infected or not. Pseudo-labels are inferred from the class of the volume $V$ containing cuboid $c$. Ignoring the infection severity at this stage, we are able to infer that if a cuboid $c \in V$ and $V$ is negative for infection, then $c$ is negative. On the other hand, if $c \in V$ and $V$ is positive, then $c$ should be positive. This is formulated as a classification task and the model is trained using Cross Entropy loss. The light-weight cuboid-level classification enables the focus on local features that distinguish between normal and infected regions.

## 2.2 Volume-level classification

A Multi-Layer Perception (MLP) network is used to take the combined cuboid representations from each volume $V$ and provide an overall classification for the infection severity. Inspired by work done recently in vision Transformer models [26], we utilise 1D positional embedding for each cuboid to ensure that the MLP takes the position of the cuboid into consideration to learn inter-cuboid correspondence. The sum of each cuboid's positional embedding vector with its representation is concatenated with the volume-level representation and used to classify $V$. Furthermore, the learned embedding from the proposal generator is also passed through the MLP to provide the model with global features from $V$.

## 2.3 Weak supervision

To train the cuboid nominator model, we utilize the outputs of the cuboid-level and volume-level classifiers. As such, we use the following Binary cross-entropy function to train the Cuboid Nominator:

$$l(x_n, y_n) = y_n.\log x_n + (1 - y_n).\log(1 - x_n) \tag{1}$$

where $n \in \{1, \ldots, k\}$ and $x_n$ is the probability of the $n^{th}$ cuboid being useful as predicted by the cuboid nominator, while $y_n$ is determined using the following weakly supervised criteria:

$$y_n = \begin{cases} 1 & \text{if } V \text{ and } c_n \text{ are correctly classified.} \\ \\ \mu_i \times x_n & \begin{array}{l}\text{if } V \text{ is misclassified and } c_n \text{ was not} \\ \text{nominated or if } V \text{ is misclassified} \\ \text{and } c_n \text{ was correctly classified.}\end{array} \\ \\ \gamma_i \times x_n & \begin{array}{l}\text{if } V \text{ is correctly classified and } c_n \text{ was} \\ \text{misclassified.}\end{array} \\ \\ 0 & \text{if } V \text{ and } c_n \text{ are misclassified.} \end{cases} \tag{2}$$

where $\mu_i$ and $\gamma_i$ are the confidence increase and decrease coefficients at epoch $i$, respectively. This is built on the intuition that cuboids leading to a correct classification are likely to contain relevant visual information, which is a region of infection in the cases of positive infection cases. Conversely, cuboids that lead to the wrong overall classification are less likely to contain relevant information. Intuitively, the criteria should also decrease the probability of cuboids that are not selected when the volume is classified correctly. However, due to the large number of cuboids that fall under this category, this leads to a deflation of the total cuboid probability scores. The two middle cases in (2) allow for a transition between the two ends of the confidence spectrum. In such manner, the confidence coefficients are given by the following formulae:

$$\mu_i = \mu + \left(\frac{1}{1 + e^{-\tau_i}} - 0.5\right) \tag{3}$$

$$\gamma_i = \gamma - \left(\frac{1}{1 + e^{-\tau_i}} - 0.5\right) \tag{4}$$

where $\tau_i = \tau \times i$ and $\tau$ is an initial temperature value that controls the impact of the coefficients as the training progresses.

## 2.4 Infection clusters

Lung infections often cluster in particular areas of the lungs and include various types such as Viral Pneumonia (including COVID-19 and Influenza), Bacterial Pneumonia, Fungal Pneumonia, and Tuberculosis. In Viral Pneumonia cases, bilateral ground-glass opacities commonly appear in a clustered distribution, typically located subpleurally and predominantly at the base of the lungs. To deal with this, we propose a further step for cuboids to impact each other's probability of being picked by the cuboid nominator, as depicted in Fig 2. For this, *prior* to applying the weak supervision policy in (2), we adjust $y_n$ as follows:

$$y_n = x_n + \frac{\sum_{j=1}^{K} \epsilon_{c_n, c_j}}{K} \tag{5}$$

where $K$ is the number of cuboids in volume $V$ and $\epsilon_{c_n, c_j}$ is given by:

$$\epsilon_{c_n, c_j} = -\frac{2}{1 + e^{x_j \times \left(\frac{1}{\delta_{c_n, c_j}} - \omega\right)}} + 1 \tag{6}$$

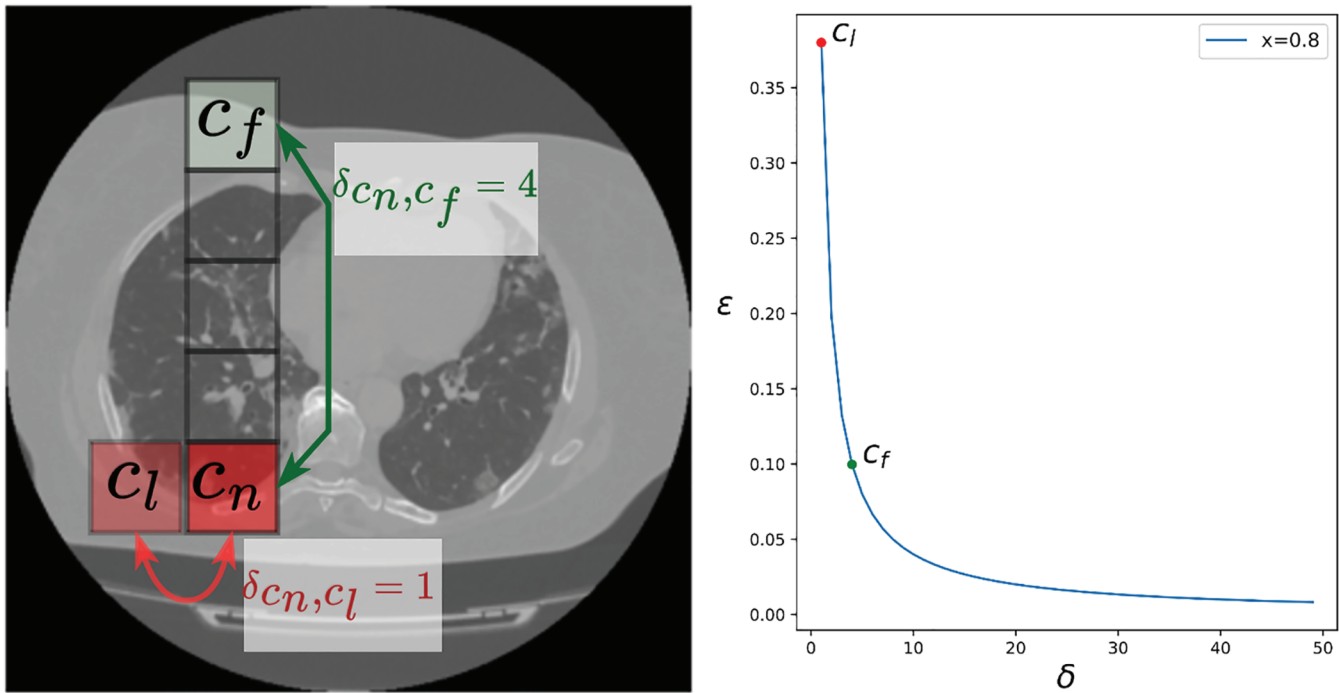

**Fig 2. An illustration of the infection clusters impact from two cuboids $c_f$ and $c_l$ on cuboid $c_n$ along the $z$ axis (Left).** The relationship between $\delta$ and $\epsilon$ for a given probability $x = 0.8$ along with the impact of $c_f$ and $c_l$ (Right).

where $\omega$ is a confidence threshold to limit the cuboids' impact after a certain limit, while $\delta_{c_n,c_j}$ is the normalized distance between cuboids $n$ and $j$ as given by:

$$\delta_{c_n,c_j} = \frac{|z_n - z_j|}{z} + \frac{|h_n - h_j|}{h} + \frac{|w_n - w_j|}{w} \tag{7}$$

To ensure that the cuboid does not have an impact on itself, we set $\frac{1}{\delta_{c_n,c_n}} = \omega$. Fig 2 illustrates an example of the infection clusters comparing the impact of two cuboids at varying distances from a given cuboid $c_n$. For an infection like Miliary Pneumonia, which presents in small pulmonary nodules scattered across the lungs, we can set $\omega$ to a very small value to limit the cuboid's impact and maintain a local impact.

## 3 Experimental setup

### 3.1 Datasets

We use two datasets, HUST [7] and MosMed [27], collected from China and Russia, respectively. This is to evaluate our proposed method on diverse data and show its ability to perform well compared to existing methods. We match the infection severity labels between the two datasets. The HUST dataset followed the guidance of the Chinese National Health Commission [28] in labelling the COVID-19 infection severities, which employs a combination of clinical features and CT observations. Similarly, MosMed uses a combination of clinical features along with visual features from the patient's CT scan to determine the severity as summarized in [27]. In both datasets, each patient is diagnosed as one of five classes: control, mild, medium, severe, and critical. With slight variations between the class-determining

criteria in terms of clinical features, the CT visual representation is more consistent, where a mild case is determined by little to no signs of pneumonia in the CT scan. A medium case is determined by some signs of pneumonia (25 – 50% coverage as defined in MosMed), while a severe case would have signs of more than 50% pulmonary parenchymal involvement. Finally, a critical case is determined when there is respiratory failure, shock, and more than 75% coverage.

We use chest CT images from a total of 2,631 patients, 1,345 are part of the HUST dataset collected from two hospitals in China. The remainder 1,110 patients are part of the MosMed dataset collected from municipal hospitals in Moscow, Russia. The average age of patients in the HUST dataset is 56.6 years, with an average 51.1 years in the control class [7]. There is a subtle difference between the average age for different severities, with older age groups being more susceptible to develop more severe infections. As for the MosMed dataset, the median age is 47 years, with ages ranging from 18 to 97 years. The HUST dataset contains 249 control, 222 mild, 434 moderate, 131 severe, and 27 critical cases, while the remaining suspected but unconfirmed cases. The MosMed dataset contains 254 control, 684 mild, 125 moderate, 45 severe, and 2 critical cases. Due to the class imbalance challenge between the control, mild, medium, severe and critical samples, Ning et al. [7] combined mild and medium cases as Type I class and severe and critically ill cases as Type II class.

We follow four strategies in splitting the datasets: (a) In a similar approach to Ning et al. [7], for the HUST dataset, we combine the mild and medium cases to form a single class (Type I) and we combine the severe and critical cases to form another class (Type II), while the control class remains independent. In this split, we use the same cross-site approach to split the training and testing data, with the training set being from one hospital and the test set from another hospital. (b) We follow the same approach in merging the classes from the MosMed dataset into 3 classes, while splitting the MosMed dataset randomly into 80% and 20% splits for training and testing, respectively. (c) We combine the HUST and MosMed datasets by aligning their classes and we also use the 3-class approach. We conduct a 5-fold cross-validation split for this with 80% for training and 20% for testing. (d) The MosMed and HUST datasets are combined while keeping the 5 severity classes, where we also conduct 5-fold cross-validation with 80% training and 20% testing splits from both MosMed and HUST datasets. In all settings, we omit the suspected class from the HUST dataset, because they do not have a definitive laboratory confirmation of COVID-19 at the time of enrolment. Both splits are carried out at patient-level, with a single scan per patient, without any follow-up scans. Furthermore, patients from both datasets are enrolled from different hospitals, and from different countries in the case of HUST and MosMed datasets, with minimal likelihood of double patient enrolment between datasets.

A subset of 50 CT scans belonging to mild cases from the MosMed dataset [27] has been annotated by experts of the Research and Practical Clinical Center for Diagnostics and Telemedicine Technologies of the Moscow Healthcare Department. The annotators marked ground-glass opacifications and regions of consolidation in each of the 50 scans. This subset is used to evaluate the explainability of the model by measuring infection coverage.

## 3.2 Image preprocessing

The CT images from the two datasets are resampled to a unified isotropic resolution with $1mm \times 1mm \times 1mm$ voxel size. All images are also resized to $64 \times 128 \times 128$ voxels using first-order spline interpolation, while the Hounsfield units of every voxel are clipped to values between −1250 and 200 and then normalized. To enable better generalization and to prevent

overfitting, we apply on-the-fly volume augmentation with random scaling between 0.9 and 1.1. We finally apply a random lateral flip with a probability of 0.5.

### 3.3 Experiments

We implement an end-to-end training approach for the three components of the pipeline, the cuboid nominator, the cuboid-level classifier and the volume-level classifier. While the cuboid nominator is trained to reduce the Binary Cross Entropy loss in (1), the cuboid-level and volume-level classifiers are both trained using the Cross Entropy (CE) criterion, where soft labels for the cuboids are deduced from the volume's label. Regardless of the number of classes for the volume, we observe that training the cuboid-level classifier using binary labels (infected or not) achieves better overall performance. Furthermore, weights are applied to the CE criterion at the volume-level to deal with the class imbalance problem. This is applied to the 3-class and 5-class tasks. Stratified sampling is also used for each batch of volumes to better deal with the class imbalance challenges.

We empirically chose the 3D-DenseNet-121 model [29] as the cuboid nominator with a penultimate fully-connected layer of size 1024, used as the volume-level representation. We choose a cuboid size of $8 \times 16 \times 16$ as it is small enough to provide a good localisation precision but also large enough to contain significant features of infection to be classified. A shallow 3D CNN model is used for the cuboid-level classification, with two 3D convolutional layers of $(2 \times 2 \times 2)$ and $(1 \times 1 \times 1)$ kernel sizes along with a stride of $(1,2,2)$ for each layer. Following each convolutional layer, we use a max pooling layer with a window size of $(2 \times 2 \times 2)$. Then, three fully-connected layers (2048, 1024 and 128 neurons) are used before the model ends with an output size of 2 to determine whether the cuboid is infected or not. The output of the penultimate layer with 128 neurons is considered cuboid-level representation. We then use an MLP model for the volume-level classification, where the 128 cuboid representation is added to the positional embeddings for the selected cuboids. The result is then concatenated with the 1024 volume representation from the cuboid nominator and passed through two fully connected layers of 512 and 128, ending with an output size equivalent to the number of classes in the task. The cuboid-level and volume-level classifiers' design ensures minimal added computational complexity over the vanilla 3D-DenseNet-121 model, with less than 0.1% of added floating point operations per second at inference time.

All models in the pipeline are trained using the ADAM optimizer [30] with 0.9 and 0.999 values for $\beta_1$ and $\beta_2$ respectively. All models are trained with a starting learning rate of 0.001 and a reduce-on-plateau learning rate scheduler for each model to reduce the learning rate by a factor of 0.1 after 25 epochs without reduction in the respective model's training loss, with a lower bound of $1 \times 10^{-6}$. Each experiment was run on a single NVIDIA V100 GPU.

## 4 Results and analysis

We evaluate our method using: (a) The HUST test subset containing 395 patients collected from a different hospital to that of the training data [7]. (b) A 20% random split of the MosMed dataset. To help better assess generalizability, we use the HUST test set collected by Ning et al. [7] from a separate hospital to that of the training set. We can also use this to address robustness and reliability challenges, which are one of the common pitfalls for data-driven methods developed for diagnosis and prognostication [21]. Furthermore, we focus on the macro averaged $F_1$ measure because of the class imbalance between infection severity classes since accuracy could be misleading of the true model performance. Through those different test settings, we can compare our method to other existing methods in terms of generalization, diagnosis performance, and dealing with class imbalance.

In Table 1, we compare the performance of our approach with that of the baseline 3D-DenseNet-121 model, slice-wise VGG-16 model [7], and the method proposed by Han et al.'s [8] as a weakly supervised multi-instance learning approach (AD3D-MIL). The baseline of 3D-DenseNet-121 model is the same architecture used as the backbone cuboid nominator model for our approach. It is trained from scratch using the same image preprocessing techniques described in Sect 3.2. On the other hand, the performance measures for the slice-wise VGG-16 model, including $F_1$, are calculated from the CT-only prediction confusion matrix provided in the supplementary material of [7]. Finally, we use the AD3D-MIL code provided by Han et al. [8] to train and test their weakly supervised multi-instance learning approach. A U-Net model is first used to segment the lungs from the CT scans, we then train the attention-based multi-instance 3D deep model to classify infection severity. We publicly release the adapted AD3D-MIL code for weakly supervised multiple instance learning for COVID-19 infection severity classification (https://github.com/ibrahimalmakky/AD3DMIL-InfectionSeverity).

**Generalization.** In Table 1, we observe that our approach achieves significant gain in $F_1$ measure on the cross-site HUST data split. In comparison, other approaches struggle with the domain shift between the train and test sets. We hypothesize that our model's increased ability to cope with such distribution shifts is due to the combination of high-level (i.e., volume-level) and low-level (i.e., cuboid-level) features learned using the cuboid nominator and classifier, respectively. On the other hand, AD3D-MIL suffers from overfitting on the training data, without being able to generalize well on the test set. This results in a significantly lower performance when compared to the test results on the MosMed dataset.

**Infection severity.** Class imbalance poses a significant challenge to accurate infection diagnosis from chest CT scans. The effect of this is evident in Table 1, where high Recall rates are reported for AD3D-MIL and Slice-wise VGG-16 models. On the other hand, our approach is able to address this issue and achieve state-of-the-art $F_1$ score on the HUST and MosMed datasets. Learning from cuboid-level features allows the model to better distinguish between infection severities, with minimal confusion between control and Type II classes in both HUST and MosMed cases (Fig 3). For MosMed, the confusion is mostly concentrated between control and Type I and between Type I and Type II. Interestingly, the confusion between Type I and control is significantly lower in the HUST dataset and between control and Type I cases (Fig 3). The former is likely due to the better representation of Type II severity samples in the HUST dataset, when compared to MosMed.

**Table 1**. Classification results on the following two settings: (1) Three combined infection severity classes from the the HUST dataset [7]: control, Type I (Mild and medium cases), and Type II (Severe and critical cases). (2) The same three classes, but from the combined HUST and MosMed datasets. [†]Based on the confusion matrix provided by the authors in the supplementary materials of [7]. [‡]Based on 5 runs with different random seeds to overcome the non-deterministic 3D convolutional.

| Dataset | Method | Precision | Recall | Accuracy | F1 Score |
|---|---|---|---|---|---|
| HUST [7] | AD3D-MIL [8] | 0.46 | 0.41 | 0.54 | 0.41 |
| | Slice-wise VGG-16 [7][†] | 0.55 | **0.75** | 0.68 | 0.57 |
| | 3D-DenseNet-121[‡] | 0.60 ± 0.03 | 0.59 ± 0.03 | 0.62 ± 0.03 | 0.57 ± 0.02 |
| | **Ours - DenseNet-121 Backbone**[‡] | **0.62** ± 0.02 | 0.61 ± 0.03 | **0.64** ± 0.01 | **0.60** ± 0.03 |
| MosMed [27] | AD3D-MIL [8] | 0.59 | **0.72** | **0.85** | 0.62 |
| | 3D-DenseNet-121[‡] | 0.61 ± 0.04 | 0.59 ± 0.04 | 0.75 ± 0.03 | 0.60 ± 0.01 |
| | **Ours - DenseNet-121 Backbone**[‡] | **0.65** ± 0.05 | 0.65 ± 0.04 | 0.73 ± 0.03 | **0.63** ± 0.01 |

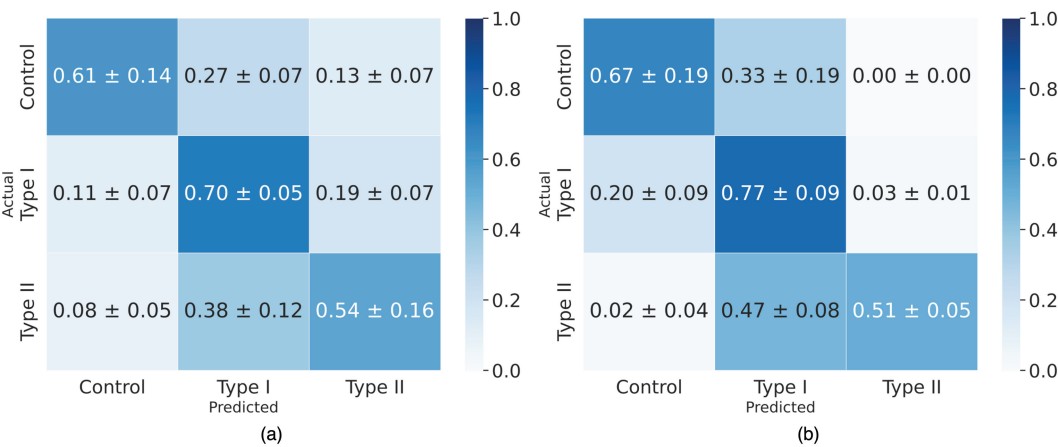

**Fig 3. Confusion matrices for our method on the (a) HUST [7] and (b) MosMed [27] datasets.**

# 5 Discussion

## 5.1 Explainability

Explainability is an integral part of diagnosis methods, which could enable them to be considered for adoption in clinical practice [21]. The explainability of our approach stems from the weakly-supervised cuboid nominator, which is learning to identify "useful" cuboids for the classification task. The resulting cuboids can provide an idea of where the infection is located within the volume as demonstrated in Fig 4. To quantify explainability, we measure the cuboids' coverage rate $C$ for the infected area using the subset of labelled volumes provided with the MosMed test data. We define $C$ as the percentage of infection covered by the proposed cuboids to the overall infection in the volume. It is important to note that the aim is to achieve a coverage level that provides an explainable result, but does not necessarily compete with the performance of a supervised infection segmentation model.

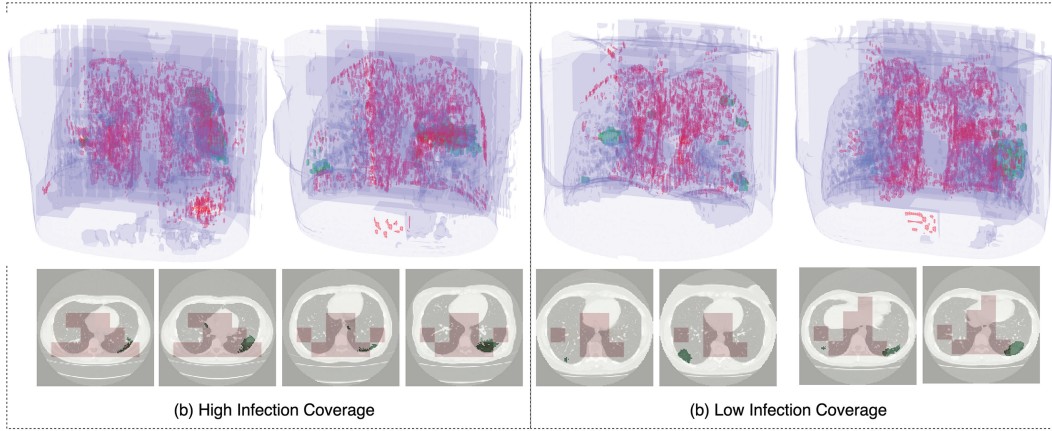

**Fig 4. Explainability visualization as a result of highlighting the contents of the selected cuboids (red) compared to manually annotated infection mask (green).** (a) shows two validation samples with the highest infection coverage score while (b) shows the two validation samples with the lowest infection coverage score.

Fig 5 illustrates the increase in the infection coverage for training and validation subsets, demonstrating that the cuboid nominator is learning to pick cuboids containing infection. Our approach can achieve an infection coverage of around 65% on unseen data (Fig 5) when setting $k$ number of cuboids to be proposed by the cuboid nominator to 64 out of 512 possible cuboids. This infection coverage rate along with the visualizations in Fig 4 highlight the significance of our approach towards clinical adoption. Our method can provide clinicians with an infection severity classification along with areas to focus on in the CT scan, thus accelerating the diagnosis process and enabling wider adoption when compared with black-box methods. All annotated CT volumes used for validation are from the subset of annotated CT scans from the MosMed dataset. All annotated samples belong to the mild infection class, which has small lesion coverage making it more challenging.

In Fig 5, we analyse the impact of varying the $k$ number of selected cuboids on the $F_1$ measure and the infection coverage. Intuitively, we observe an increase in infection coverage as we increase $k$, until we reach a saturation level at $k = 64$, where we also achieve a maximum $F_1$ score. We also observe an unstable infection coverage score and $F_1$ measure with $k = 80$, caused by the large number of cuboids shifting within the volume, especially at the beginning of training. As for determining the cuboid dimensions; a smaller cuboid size can provide higher localisation resolution, but a very small size would make it challenging for the cuboid-level classifier to determine whether the cuboid contains infection or not. We experimentally tested varying the cuboid dimensions, through which we noticed that the cuboid proximity factor allows us to use small cuboid dimensions to achieve higher infection coverage while maintaining classification performance.

## 5.2 Ablation study

**5.2.1 Combined datasets.** In Table 2, we test our method using a five-fold cross-validation split of our combination of the HUST [7] and MosMed [27] datasets. The prediction for the combination of the HUST and MosMed datasets shows a compromise between the two individual dataset predictions. The increase in the number of samples leads to increased performance for all methods when combining the two datasets, when it comes to

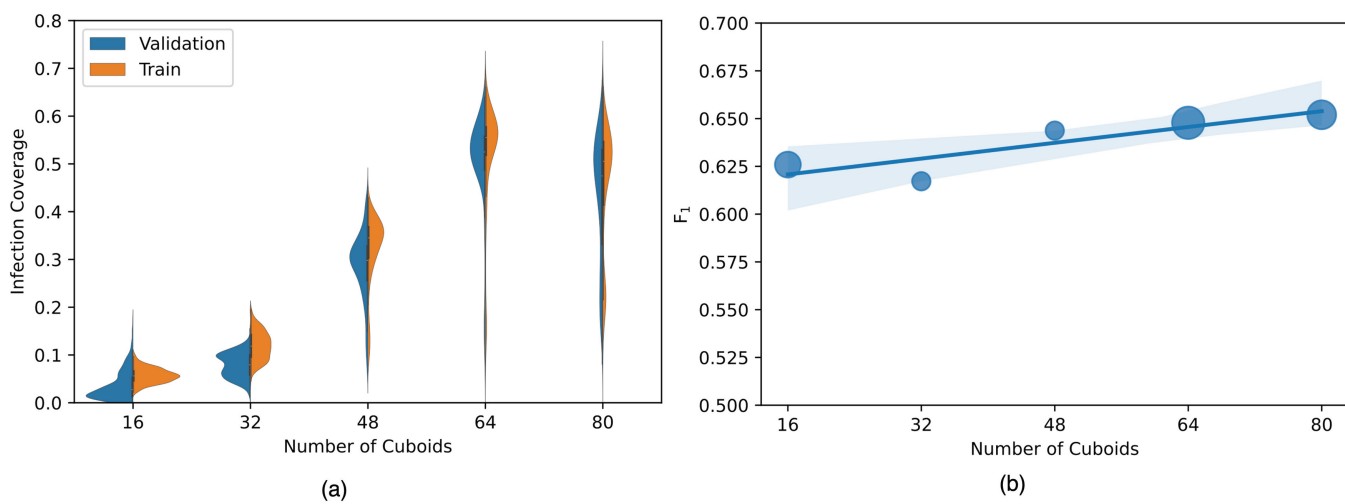

(a)

(b)

**Fig 5. The impact of varying the $k$ number of nominated cuboids on: (a) the training and validation infection coverage rate during the training process.** (b) The validation $F_1$ measure, with the size of the dots indicating how early in the training process, the best $F_1$ measure was achieved.

**Table 2**. Classification results on the following three settings: (1) Three combined infection severity classes from the the HUST dataset [7]: control, Type I (Mild and medium cases), and Type II (Severe and critical cases). (2) The same three classes, but from the combined HUST and MosMed datasets. (3) All five classes, control, mild, medium, severe and critical, from the same split of the combined HUST and MosMed datasets.

| Dataset | # Classes | Method | Precision | Recall | Accuracy | F1 Score |
|---|---|---|---|---|---|---|
| MosMed [27]+HUST [7] | 5 | AD3D-MIL [8] | 0.44 ± 0.06 | 0.44 ± 0.04 | 0.63 ± 0.04 | 0.4196 ± 0.05 |
| | | **Ours - DenseNet-121 Backbone** | **0.55** ± 0.04 | **0.54** ± 0.04 | **0.63** ± 0.02 | **0.52** ± 0.03 |
| MosMed [27]+HUST [7] | 3 | AD3D-MIL [8] | 0.63 ± 0.03 | 0.61 ± 0.03 | **0.74** ± 0.01 | 0.61 ± 0.04 |
| | | **Ours - DenseNet-121 Backbone** | **0.64** ± 0.01 | **0.66** ± 0.02 | 0.70 ± 0.02 | **0.64** ± 0.004 |

the three-class setting. We also test on the five-class severity setting, which is significantly more imbalanced than the three-class setting. The impact of this is clear, with significantly lower $F_1$ measures reported from all methods compared to the three-class setting. This is likely due to the increased task complexity in learning the low and high-level features from two datasets with aligned infection severity classes. In Fig 7, we observe the highest confusion in severe cases that are misclassified as either medium or critical cases, with a tendency to go for the less severe ones. Even though zero control and mild cases were classified as critical, there were some critical cases misclassified as control, which is likely caused by the class imbalance, where the critical class has a minimal number of samples in both datasets. Furthermore, even though the five-class classification task proved to be more challenging and resulted in lower $F_1$ score, our cuboid nominator is still able to achieve similar infection coverage to that of the three class classification task (Fig 6).

**5.2.2 Infection clusters.** To understand the impact of our approach from Sect 2.4 in allowing local influence of high probability cuboids on nearby cuboids, we experiment with activating this proximity factor at later epochs rather than starting it at the first epoch. Fig 6 shows that some delay (20 epochs) in activating the proximity factor between cuboids results in higher infection coverage, but lower $F_1$. However, the activation of the proximity factor from the first epoch yields the highest $F_1$ measure.

**5.2.3 Volume-level features.** Our approach employs a combination of high and low-level features for the volume classification, with a combination of features extracted by the cuboid nominator along with features extracted from cuboids. To analyse the contribution

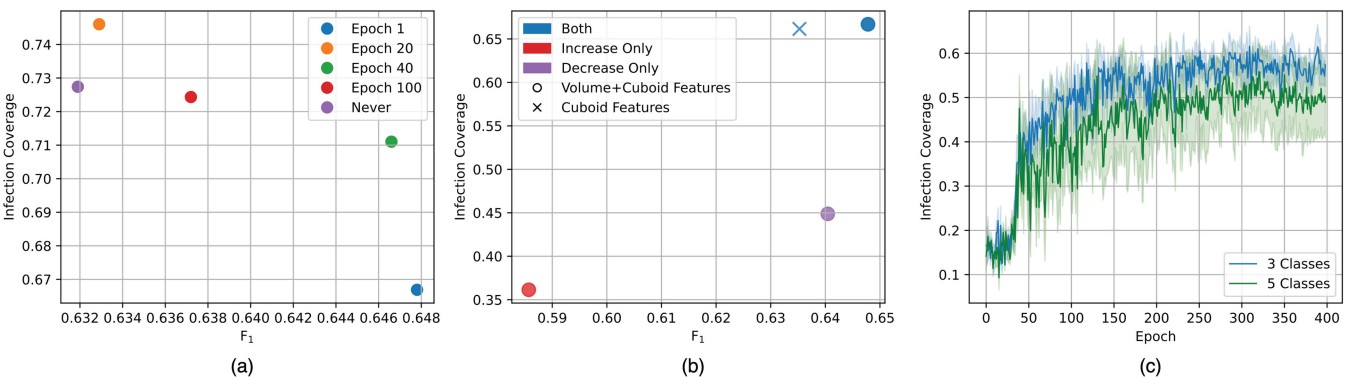

(a)                          (b)                          (c)

**Fig 6. (a) The impact of changing the epoch at which the proximity factor for infection clusters is activated on the validation infection coverage and $F_1$ measure.** (b) The ablation results when removing the influence of the increase and decrease coefficients from (2) on the validation infection coverage as well as the impact of removing the volume-level features. (c) Comparison between the validation infection coverage pattern during training of the three and five class settings.

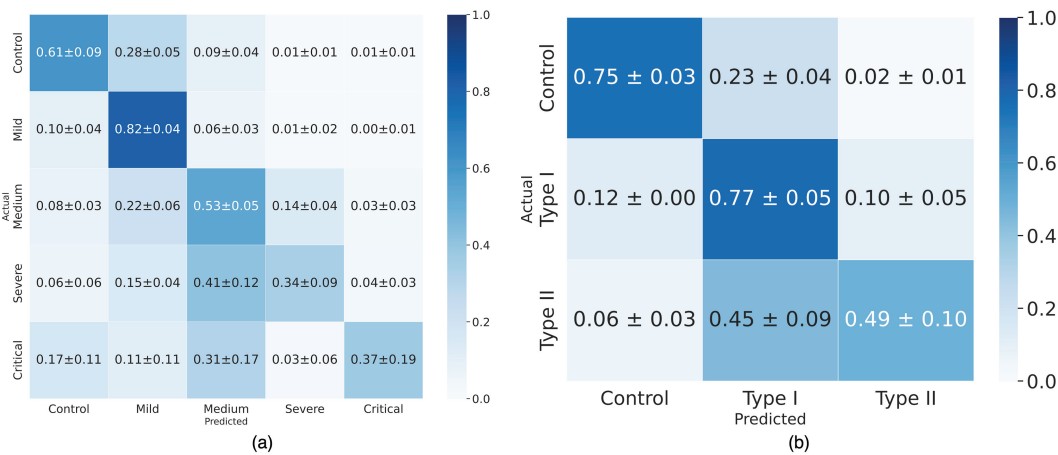

**Fig 7. The confusion matrix with mean and standard deviations for the five (a) and three (b) classes in the HUST, MosMed dataset combination when testing our proposed approach using 5-fold cross validation.**

of the volume-level features extracted by the cuboid nominator, we remove the concatenation between the cuboid features and the volume-level features. The results of this ablation are demonstrated in Fig 6, where the contribution of volume-level features is clear, with an overall higher $F_1$ and faster convergence towards the highest performance. This highlights the significance of inputting volume-level features along with cuboid-level features to the volume-level classifier, where the volume-level features provide a general outlook of the lesion coverage in the volume while the cuboid-level features provide details of the low-level pattern of infection.

**5.2.4 Weak supervision policy.** Our weak supervision policy defined in (2) balances between increasing confidence in cuboids likely to be useful for the classification task while doing the opposite for cuboids that are unlikely to be "useful" (i.e., cuboids without discriminative features). In such manner, we employ confidence increase and decrease factors, $\mu_i$ and $\gamma_i$ respectively. To understand the impact those factors have on the weakly supervised training process, we experiment with removing each of them independently and compare that result to the validation results we achieve with both of them in use (Fig 7). It is clear that both factors employed together achieve the best performance and infection coverage, but we can also observe that the decrease coefficient has a higher impact on the effectiveness of the weak supervision policy. This highlights the importance of decreasing confidence in misclassified cuboids, which in turn highlights the importance of low-level cuboid features on the overall volume classification.

# 6 Conclusions and future work

In this work, we propose an end-to-end weakly supervised explainability-focused approach for COVID-19 infection severity classification. Composed of a cuboid nominator, a cuboid classifier, and a volume-level classifier, we demonstrate state-of-the-art infection severity classification for COVID-19 patients along with interpretable results to support clinicians. The strength of our approach stems from its ability to learn from cuboid-level features that resemble the more intricate lesion features as well as the volume-level lesion coverage. The added explainability takes deep learning methods a step forward towards clinical adoption, while assisting clinicians in accelerating the diagnosis process.

The findings of this study point toward the potential to enhance clinical workflows by providing an interpretable, data-driven approach for assessing infection severity from chest CT scans. By incorporating data from multiple centers across different countries, with varying diagnostic guidelines, this work highlights the generalizability of the proposed method in diverse healthcare contexts. Notably, the explainability built into the model offers visual cues that can support clinical understanding and oversight, contributing to more informed decision-making. While further validation and integration steps would be required before any clinical deployment, the demonstrated performance and transparency mark a promising step toward developing clinically relevant AI-based tools. Moreover, the approach can accelerate the diagnostic process by helping clinicians rapidly identify and localize infection regions. This aspect could be especially valuable in high-demand settings where timely intervention is critical to improving patient outcomes.

The limitations of this work are mainly related to the focus on COVID-19 for lung infection coverage without the inclusion of other respiratory conditions. This is mainly due to the lack of data to enable the training and assessment of such a model, where the COVID-19 use case provided a good starting step. While our explainability evaluation demonstrates encouraging infection coverage results, it was conducted on a relatively small annotated subset of mild cases from the MosMed dataset. Broader validation across moderate, severe, and critical cases is therefore necessary to fully establish the clinical reliability of the proposed explainability framework. In the future, we see great potential in extending our approach to include multimodal inputs with added clinical features for volume-level classification. This would require the extension of explainability methods to involve electronic health records. Finally, deeper analysis into the automated localization of subvisual lesions in COVID-19 survivors could be an interesting area to explore as a result of this work.

## Author contributions

**Conceptualization:** Ibrahim Almakky, Mohammad Yaqub.

**Formal analysis:** Ibrahim Almakky.

**Investigation:** Ibrahim Almakky.

**Methodology:** Ibrahim Almakky, Mohammad Yaqub.

**Software:** Ibrahim Almakky.

**Validation:** Ibrahim Almakky.

**Writing – original draft:** Ibrahim Almakky.

**Writing – review & editing:** Ibrahim Almakky, Mohammad Yaqub.

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
