## [Decision Letter · Decision Letter 0]

12 Jan 2025

PONE-D-24-51803Weakly-Supervised explainable infection severity classification from chest CT scansPLOS ONE

Dear Dr. Almakky,

Thank you for submitting your manuscript to PLOS ONE. After careful consideration, we feel that it has merit but does not fully meet PLOS ONE’s publication criteria as it currently stands. Therefore, we invite you to submit a revised version of the manuscript that addresses the points raised during the review process.

We look forward to receiving your revised manuscript.

Kind regards,

Asadullah Shaikh, Ph.D.

Academic Editor

PLOS ONE

Journal Requirements:

Reviewers' comments:

Reviewer's Responses to Questions

**Comments to the Author**

1. Is the manuscript technically sound, and do the data support the conclusions?

Reviewer #1: Yes

Reviewer #2: Yes

Reviewer #3: Yes

2. Has the statistical analysis been performed appropriately and rigorously? 

Reviewer #1: Yes

Reviewer #2: Yes

Reviewer #3: Yes

3. Have the authors made all data underlying the findings in their manuscript fully available?

Reviewer #1: Yes

Reviewer #2: Yes

Reviewer #3: Yes

4. Is the manuscript presented in an intelligible fashion and written in standard English?

Reviewer #1: Yes

Reviewer #2: Yes

Reviewer #3: Yes

5. Review Comments to the Author

Reviewer #1: The authors present a weakly-supervised machine learning system to classify COVID-19 infection severity from chest CT scans. The paper is very well written, all steps of the classification pipeline are fully understandable. I thus have no comments for improvement.

Reviewer #2: This paper presents a weakly supervised classification method for how to categorize infection severity from chest CT scan images and provide clinicians with interpretable results.

1. Weakly supervised classification method:

The paper proposes a weakly supervised training method to train a Cuboid Nominator by combining the outputs of volume-level and cuboid-level classifiers. This approach avoids the strong reliance on labeled data and significantly reduces the need for manual labeling by training through pseudo-labeling and self-supervised learning.

2. Cuboid Nominator:

A 3D CNN-based cuboid nominator model is proposed, which is able to select “useful” cuboids from the input CT volume data. By extracting meaningful local regions from the overall volume data, this method focuses more on the extraction of local features than traditional methods, which helps to improve the accuracy of classification.

3. Cube-level classification:

The paper further proposes a small 3D CNN model specifically for classifying nominated cubes.By inferring whether each cube is an infected region through pseudo-labeling, this refined classification strategy helps to extract more precise features and improve the overall classification.

4. Introduction of Infection Clusters:

The paper proposes that lung infections are usually manifested as a cluster of neighboring cubes, and therefore a mechanism is devised such that the classification of each cube not only depends on itself, but also interacts with the features of the surrounding cubes. This approach takes into account the spatial correlation of the infection region and can effectively improve the classification performance, especially when dealing with data with spatial dependencies.

5. Volume-level classifier::

A volume-level classifier based on a multilayer perceptual machine (MLP) is introduced, which can better understand the spatial relationship between cubes by combining the representation of each cube and its positional encoding, thus improving the accuracy of classification.

The shortcoming include:

1. Higher model complexity:

The method introduces multiple model components (cube nominator, cube classifier, volume level classifier, etc.), and although the design of these components helps to improve the model performance, it also brings higher computational cost and model complexity. A large amount of training data and computational resources are required, which may be difficult to implement in some resource-limited environments.

2. Limitations of the infection cluster model:

The assumption for the construction of infection clusters is based on the assumption that infected regions are spatially correlated, but this assumption is not always applicable to all types of infections, especially since some infections may exhibit a non-clustered morphology. Therefore, this approach may not be applicable in some special cases.

3. Dataset limitations:

The paper mainly used two datasets, and although these datasets are from different countries and hospitals, there may still be geographical and clinical practice differences, which may affect the model's ability to generalize.

These shortcomings provide directions for future research, helping to enhance the model's practicality and effectiveness.

References

Ji Z, Mu J, Liu J, et al. ASD-Net: a novel U-Net based asymmetric spatial-channel convolution network for precise kidney and kidney tumor image segmentation[J]. Medical & Biological Engineering & Computing, 2024: 1-15. https://doi.org/10.1007/s11517-024-03025-y

Ji Z, Zhao J, Liu J, et al. ELCT-YOLO: an efficient one-stage model for automatic lung tumor detection based on CT images[J]. Mathematics, 2023, 11(10): 2344. https://doi.org/10.3390/math11102344

* Your results need to be compared with two papers, and the analysis can be conducted in the Introduction section.

Reviewer #3: 1. Authors are required to clearly highlight how the proposed weakly-supervised approach contributes to clinical adoption.

2. The strategies used to handle class imbalance should be explained more with clarity.

3. Expand the comparison with other methods beyond the F1 score and accuracy metrics.

4. Include additional insights into the model’s generalizability by evaluating it on other respiratory conditions, as the study currently focuses on COVID-19.

5. Provide a more detailed explanation of critical steps, such as the "cuboid nomination" process and its role in infection localization.

6. Expand the discussion on limitations and future work, particularly the potential of integrating multimodal data.

7. Improve the clarity of visualizations especially fig 5 and 7.

8. Revise conclusion by highlighting the key take aways from the paper.

6. PLOS authors have the option to publish the peer review history of their article (what does this mean?). If published, this will include your full peer review and any attached files.

Reviewer #1: No

Reviewer #2: No

Reviewer #3: No

---

## [Author Response · Author response to Decision Letter 1]

24 Mar 2025

We thank the reviewers and the editor for their time and constructive feedback. We attach a detailed document containing the reviewers' comments along with our responses and the changes we have made in the paper.

---

## [Decision Letter · Decision Letter 1]

15 Apr 2025

PONE-D-24-51803R1Weakly-Supervised explainable infection severity classification from chest CT scansPLOS ONE

Dear Dr. Almakky,

Thank you for submitting your manuscript to PLOS ONE. After careful consideration, we feel that it has merit but does not fully meet PLOS ONE’s publication criteria as it currently stands. Therefore, we invite you to submit a revised version of the manuscript that addresses the points raised during the review process.

We look forward to receiving your revised manuscript.

Kind regards,

Asadullah Shaikh, Ph.D.

Academic Editor

PLOS ONE

Journal Requirements:

Reviewers' comments:

Reviewer's Responses to Questions

**Comments to the Author**

1. If the authors have adequately addressed your comments raised in a previous round of review and you feel that this manuscript is now acceptable for publication, you may indicate that here to bypass the “Comments to the Author” section, enter your conflict of interest statement in the “Confidential to Editor” section, and submit your "Accept" recommendation.

Reviewer #3: (No Response)

2. Is the manuscript technically sound, and do the data support the conclusions?

Reviewer #3: Yes

3. Has the statistical analysis been performed appropriately and rigorously? 

Reviewer #3: Yes

4. Have the authors made all data underlying the findings in their manuscript fully available?

Reviewer #3: Yes

5. Is the manuscript presented in an intelligible fashion and written in standard English?

Reviewer #3: Yes

6. Review Comments to the Author

Reviewer #3: The method is technically sound and shows strong performance across datasets.

Minor suggestions:

1, Consider briefly discussing how this method could fit into clinical workflows.

2, Visuals like Fig. 5 and Fig. 7 could still benefit from slight clarity improvements.

Overall, a solid paper just a few small refinements needed.

7. PLOS authors have the option to publish the peer review history of their article (what does this mean?). If published, this will include your full peer review and any attached files.

Reviewer #3: **Yes: **Ashish Shiwlani

---

## [Author Response · Author response to Decision Letter 2]

18 Jun 2025

We thank the reviewers and academic editor for their efforts in reviewing this work and for their constructive feedback. We have attached a document containing our response and addressing each point raised by the reviewers, providing comments and referring them to changes we have made in the revised manuscript.

---

## [Decision Letter · Decision Letter 2]

7 Jul 2025

PONE-D-24-51803R2Weakly-Supervised explainable infection severity classification from chest CT scansPLOS ONE

Dear Dr. Almakky,

Thank you for submitting your manuscript to PLOS ONE. After careful consideration, we feel that it has merit but does not fully meet PLOS ONE’s publication criteria as it currently stands. Therefore, we invite you to submit a revised version of the manuscript that addresses the points raised during the review process.

We look forward to receiving your revised manuscript.

Kind regards,

Asadullah Shaikh, Ph.D.

Academic Editor

PLOS ONE

Journal Requirements:

Reviewers' comments:

Reviewer's Responses to Questions

**Comments to the Author**

1. If the authors have adequately addressed your comments raised in a previous round of review and you feel that this manuscript is now acceptable for publication, you may indicate that here to bypass the “Comments to the Author” section, enter your conflict of interest statement in the “Confidential to Editor” section, and submit your "Accept" recommendation.

Reviewer #3: (No Response)

2. Is the manuscript technically sound, and do the data support the conclusions?

Reviewer #3: Yes

3. Has the statistical analysis been performed appropriately and rigorously? 

Reviewer #3: Yes

4. Have the authors made all data underlying the findings in their manuscript fully available?

Reviewer #3: Yes

5. Is the manuscript presented in an intelligible fashion and written in standard English?

Reviewer #3: Yes

6. Review Comments to the Author

Reviewer #3: the paper is very well written, I have minor recommendations that would improve the paper quality much better.

Please provide a bit more detail on your model architecture and training settings to help with reproducibility also

Consider elaborating on the limitations and possible real world challenges of your approach and the last one to add few figures could be improved for clarity.

7. PLOS authors have the option to publish the peer review history of their article (what does this mean?). If published, this will include your full peer review and any attached files.

Reviewer #3: **Yes: **Ashish Shiwlani

---

## [Author Response · Author response to Decision Letter 3]

15 Sep 2025

We thank the Reviewer for their comments about our paper. Below are our responses to their minor suggestions:

1. We have added the Github repo to the code implementation and the hyperparamteres used to achieve each metric. We have provided the link in the paper in footnote of page 3.

2. We added an extension to the limitation description in the conclusion that describes the impact of using the MosMed dataset subset on the explainability validation. As for the clarity of the figures, we recomment to view the the full-resolution figures using the download links at the top right hand corner of the figure pages.

---

## [Decision Letter · Decision Letter 3]

28 Sep 2025

Weakly-Supervised explainable infection severity classification from chest CT scans

PONE-D-24-51803R3

Dear Dr. Almakky,

We’re pleased to inform you that your manuscript has been judged scientifically suitable for publication and will be formally accepted for publication once it meets all outstanding technical requirements.

Kind regards,

Asadullah Shaikh, Ph.D.

Academic Editor

PLOS ONE

Additional Editor Comments (optional):

Reviewers' comments:

Reviewer's Responses to Questions

**Comments to the Author**

1. If the authors have adequately addressed your comments raised in a previous round of review and you feel that this manuscript is now acceptable for publication, you may indicate that here to bypass the “Comments to the Author” section, enter your conflict of interest statement in the “Confidential to Editor” section, and submit your "Accept" recommendation.

Reviewer #3: All comments have been addressed

2. Is the manuscript technically sound, and do the data support the conclusions?

Reviewer #3: Yes

3. Has the statistical analysis been performed appropriately and rigorously? 

Reviewer #3: Yes

4. Have the authors made all data underlying the findings in their manuscript fully available?

Reviewer #3: Yes

5. Is the manuscript presented in an intelligible fashion and written in standard English?

Reviewer #3: Yes

6. Review Comments to the Author

Reviewer #3: (No Response)

7. PLOS authors have the option to publish the peer review history of their article (what does this mean?). If published, this will include your full peer review and any attached files.

Reviewer #3: **Yes: **Ashish Shiwlani

---

## [Editor Report · Acceptance letter]

PONE-D-24-51803R3

PLOS ONE

Dear Dr. Almakky,

I'm pleased to inform you that your manuscript has been deemed suitable for publication in PLOS ONE. Congratulations! Your manuscript is now being handed over to our production team.

Kind regards,

on behalf of

Prof. Asadullah Shaikh

Academic Editor

PLOS ONE